# Sparse Variational Inference:
# Bayesian Coresets from Scratch

**Trevor Campbell**
Department of Statistics
University of British Columbia
Vancouver, BC V6T 1Z4
trevor@stat.ubc.ca

**Boyan Beronov**
Department of Computer Science
University of British Columbia
Vancouver, BC V6T 1Z4
beronov@cs.ubc.ca

## Abstract

The proliferation of automated inference algorithms in Bayesian statistics has provided practitioners newfound access to fast, reproducible data analysis and powerful statistical models. Designing automated methods that are also both computationally scalable and theoretically sound, however, remains a significant challenge. Recent work on *Bayesian coresets* takes the approach of compressing the dataset before running a standard inference algorithm, providing both scalability and guarantees on posterior approximation error. But the automation of past coreset methods is limited because they depend on the availability of a reasonable coarse posterior approximation, which is difficult to specify in practice. In the present work we remove this requirement by formulating coreset construction as sparsity-constrained variational inference within an exponential family. This perspective leads to a novel construction via greedy optimization, and also provides a unifying information-geometric view of present and past methods. The proposed *Riemannian coreset construction* algorithm is fully automated, requiring no problem-specific inputs aside from the probabilistic model and dataset. In addition to being significantly easier to use than past methods, experiments demonstrate that past coreset constructions are fundamentally limited by the fixed coarse posterior approximation; in contrast, the proposed algorithm is able to continually improve the coreset, providing state-of-the-art Bayesian dataset summarization with orders-of-magnitude reduction in KL divergence to the exact posterior.

## 1   Introduction

Bayesian statistical models are powerful tools for learning from data, with the ability to encode complex hierarchical dependence and domain expertise, as well as coherently quantify uncertainty in latent parameters. In practice, however, exact Bayesian inference is typically intractable, and we must use approximate inference algorithms such as Markov chain Monte Carlo (MCMC) [1; 2, Ch. 11,12] and variational inference (VI) [3, 4]. Until recently, implementations of these methods were created on a per-model basis, requiring expert input to design the MCMC transition kernels or derive VI gradient updates. But developments in automated tools—e.g., automatic differentiation [5, 6], "black-box" gradient estimates [7], and Hamiltonian transition kernels [8, 9]—have obviated much of this expert input, greatly expanding the repertoire of Bayesian models accessible to practitioners.

In modern data analysis problems, automation alone is insufficient; inference algorithms must also be computationally scalable—to handle the ever-growing size of datasets—and provide theoretical guarantees on the quality of their output such that statistical pracitioners may confidently use them in failure-sensitive settings. Here the standard set of tools falls short. Designing correct MCMC schemes in the large-scale data setting is a challenging, problem-specific task [10–12]; and despite

recent results in asymptotic theory [13–16], it is difficult to assess the effect of the variational family on VI approximations for finite data, where a poor choice can result in severe underestimation of posterior uncertainty [17, Ch. 21]. Other scalable Bayesian inference algorithms have largely been developed by modifying standard inference algorithms to handle distributed or streaming data processing [10, 11, 18–29], which tend to have no guarantees on inferential quality and require extensive model-specific expert tuning.

Bayesian coresets ("core of a dataset") [30–32] are an alternative approach—based on the notion that large datasets often contain a significant fraction of redundant data—that summarize and sparsify the data as a preprocessing step before running a standard inference algorithm such as MCMC or VI. In contrast to other large-scale inference techniques, Bayesian coreset construction is computationally inexpensive, simple to implement, and provides theoretical guarantees relating coreset size to posterior approximation quality. However, state-of-the-art algorithms formulate coreset construction as a sparse regression problem in a Hilbert space, which involves the choice of a weighted $L^2$ inner product [31]. If left to the user, the choice of weighting distribution significantly reduces the overall automation of the approach; and current methods for finding the weighting distribution programatically are generally as expensive as posterior inference on the full dataset itself. Further, even if an appropriate inner product is specified, computing it exactly is typically intractable, requiring the use of finite-dimensional projections for approximation [31]. Although the problem in finite-dimensions can be studied using well-known techniques from sparse regression, compressed sensing, random sketching, boosting, and greedy approximation [33–51], these projections incur an unknown error in the construction process in practice, and preclude asymptotic consistency as the coreset size grows.

In this work, we provide a new formulation of coreset construction as exponential family variational inference with a sparsity constraint. The fact that coresets form a sparse subset of an exponential family is crucial in two regards. First, it enables tractable unbiased Kullback-Leibler (KL) divergence gradient estimation, which is used in the development of a novel coreset construction algorithm based on greedy optimization. In contrast to past work, this algorithm is fully automated, with no problem-specific inputs aside from the probabilistic model and dataset. Second, it provides a unifying view and strong theoretical underpinnings of both the present and past coreset constructions through Riemannian information geometry. In particular, past methods are shown to operate in a single tangent space of the coreset manifold; our experiments show that this fundamentally limits the quality of the coreset constructed with these methods. In contrast, the proposed method proceeds along the manifold towards the posterior target, and is able to continually improve its approximation. Furthermore, new relationships between the optimization objective of past approaches and the coreset posterior KL divergence are derived. The paper concludes with experiments demonstrating that, compared with past methods, *Riemannian coreset construction* is both easier to use and provides orders-of-magnitude reduction in KL divergence to the exact posterior.

## 2   Background

In the problem setting of the present paper, we are given a probability density $\pi(\theta)$ for variables $\theta \in \Theta$ that decomposes into $N$ potentials $(f_n(\theta))_{n=1}^{N}$ and a base density $\pi_0(\theta)$,

$$\pi(\theta) := \frac{1}{Z} \exp \left( \sum_{n=1}^{N} f_n(\theta) \right) \pi_0(\theta), \tag{1}$$

where $Z$ is the (unknown) normalization constant. Such distributions arise frequently in a number of scenarios: for example, in Bayesian statistical inference problems with conditionally independent data given $\theta$, the functions $f_n$ are the log-likelihood terms for the $N$ data points, $\pi_0$ is the prior density, and $\pi$ is the posterior; or in undirected graphical models, the functions $f_n$ and $\log \pi_0$ might represent $N + 1$ potentials. The algorithms and analysis in the present work are agnostic to their particular meaning, but for clarity we will focus on the setting of Bayesian inference throughout.

As it is often intractable to compute expectations under $\pi$ exactly, practitioners have turned to approximate algorithms. Markov chain Monte Carlo (MCMC) methods [1, 8, 9], which return approximate samples from $\pi$, remain the gold standard for this purpose. But since each sample typically requires at least one evaluation of a function proportional to $\pi$ with computational cost $\Theta(N)$, in the large $N$ setting it is expensive to obtain sufficiently many samples to provide high confidence in empirical estimates. To reduce the cost of MCMC, we can instead run it on a small, weighted subset

of data known as a *Bayesian coreset* [30], a concept originating from the computational geometry and optimization literature [52–57]. Let $w \in \mathbb{R}_{\geq 0}^N$ be a sparse vector of nonnegative weights such that only $M \ll N$ are nonzero, i.e. $\|w\|_0 := \sum_{n=1}^N \mathbb{1}\left[w_n > 0\right] \leq M$. Then we approximate the full log-density with a $w$-reweighted sum with normalization $Z(w) > 0$ and run MCMC on the approximation[1],

$$\pi_w(\theta) := \frac{1}{Z(w)} \exp\left(\sum_{n=1}^N w_n f_n(\theta)\right) \pi_0(\theta), \tag{2}$$

where $\pi_1 = \pi$ corresponds to the full density. If $M \ll N$, evaluating a function proportional to $\pi_w$ is much less expensive than doing so for the original $\pi$, resulting in a significant reduction in MCMC computation time. The major challenge posed by this approach, then, is to find a set of weights $w$ that renders $\pi_w$ as close as possible to $\pi$ while maintaining sparsity. Past work [31, 32] formulated this as a sparse regression problem in a Hilbert space with the $L^2(\hat{\pi})$ norm for some weighting distribution $\hat{\pi}$ and vectors[2] $g_n := (f_n - \mathbb{E}_{\hat{\pi}}\left[f_n\right])$,

$$w^\star = \operatorname*{arg\,min}_{w \in \mathbb{R}^N} \ \mathbb{E}_{\hat{\pi}}\left[\left(\sum_{n=1}^N g_n - \sum_{n=1}^N w_n g_n\right)^2\right] \quad \text{s.t. } w \geq 0, \ \|w\|_0 \leq M. \tag{3}$$

As the expectation is generally intractable to compute exactly, a Monte Carlo approximation is used in its place: taking samples $(\theta_s)_{s=1}^S \overset{\text{i.i.d.}}{\sim} \hat{\pi}$ and setting $\hat{g}_n = \sqrt{S}^{-1}\left[g_n(\theta_1) - \bar{g}_n, \ldots, g_n(\theta_S) - \bar{g}_n\right]^T \in \mathbb{R}^S$ where $\bar{g}_n = \frac{1}{S}\sum_{s=1}^S g_n(\theta_s)$ yields a linear finite-dimensional sparse regression problem in $\mathbb{R}^S$,

$$w^\star = \operatorname*{arg\,min}_{w \in \mathbb{R}^N} \ \left\|\sum_{n=1}^N \hat{g}_n - \sum_{n=1}^N w_n \hat{g}_n\right\|_2^2 \quad \text{s.t. } w \geq 0, \ \|w\|_0 \leq M, \tag{4}$$

which can be solved with sparse optimization techniques [31, 32, 34–36, 45, 46, 48, 58–60]. However, there are two drawbacks inherent to the Hilbert space formulation. First, the use of the $L^2(\hat{\pi})$ norm requires the selection of the weighting function $\hat{\pi}$, posing a barrier to the full automation of coreset construction. There is currently no guidance on how to select $\hat{\pi}$, or the effect of different choices in the literature. We show in Sections 4 and 5 that using such a fixed weighting $\hat{\pi}$ fundamentally limits the quality of coreset construction. Second, the inner products typically cannot be computed exactly, requiring a Monte Carlo approximation. This adds noise to the construction and precludes asymptotic consistency (in the sense that $\pi_w \not\to \pi_1$ as the sparsity budget $M \to \infty$). Addressing these drawbacks is the focus of the present work.

## 3 Bayesian coresets from scratch

In this section, we provide a new formulation of Bayesian coreset construction as variational inference over an exponential family with sparse natural parameters, and develop an iterative greedy algorithm for optimization.

### 3.1 Sparse exponential family variational inference

We formulate coreset construction as a sparse variational inference problem,

$$w^\star = \operatorname*{arg\,min}_{w \in \mathbb{R}^N} \ \mathrm{D_{KL}}\left(\pi_w \| \pi_1\right) \quad \text{s.t.} \quad w \geq 0, \ \|w\|_0 \leq M. \tag{5}$$

Expanding the objective and denoting expectations under $\pi_w$ as $\mathbb{E}_w$,

$$\mathrm{D_{KL}}\left(\pi_w \| \pi_1\right) = \log Z(1) - \log Z(w) - \sum_{n=1}^N (1 - w_n)\mathbb{E}_w\left[f_n(\theta)\right]. \tag{6}$$

Eq. (6) illuminates the major challenges with the variational approach posed in Eq. (5). First, the normalization constant $Z(w)$ of $\pi_w$—itself a function of the weights $w$—is unknown; typically, the form of the approximate distribution is known fully in variational inference. Second, even if the constant were known, computing the objective in Eq. (5) requires taking expectations under $\pi_w$, which is in general just as difficult as the original problem of sampling from the true posterior $\pi_1$.

Two key insights in this work address these issues and lead to both the development of a new coreset construction algorithm (Algorithm 1) and a more comprehensive understanding of the coreset construction literature (Section 4). First, the coresets form a sparse subset of an exponential family: the nonnegative weights form the natural parameter $w \in \mathbb{R}_{\geq 0}^N$, the component potentials $(f_n(\theta))_{n=1}^N$ form the sufficient statistic, $\log Z(w)$ is the log partition function, and $\pi_0$ is the base density,

$$\pi_w(\theta) := \exp\left(w^T f(\theta) - \log Z(w)\right) \pi_0(\theta) \qquad f(\theta) := \left[\begin{array}{ccc} f_1(\theta) & \ldots & f_N(\theta) \end{array}\right]^T. \quad (7)$$

Using the well-known fact that the gradient of an exponential family log-partition function is the mean of the sufficient statistic, $\mathbb{E}_w[f(\theta)] = \nabla_w \log Z(w)$, we can rewrite the optimization Eq. (5) as

$$w^\star = \underset{w \in \mathbb{R}^N}{\arg\min} \quad \log Z(1) - \log Z(w) - (1-w)^T \nabla_w \log Z(w) \quad \text{s.t.} \quad w \geq 0, \ \|w\|_0 \leq M. \quad (8)$$

Taking the gradient of this objective function and noting again that, for an exponential family, the Hessian of the log-partition function $\log Z(w)$ is the covariance of the sufficient statistic,

$$\nabla_w \mathrm{D}_{\mathrm{KL}}\left(\pi_w || \pi_1\right) = -\nabla_w^2 \log Z(w)(1-w) = -\mathrm{Cov}_w\left[f, f^T(1-w)\right], \quad (9)$$

where $\mathrm{Cov}_w$ denotes covariance under $\pi_w$. In other words, increasing the weight $w_n$ by a small amount decreases $\mathrm{D}_{\mathrm{KL}}\left(\pi_w || \pi_1\right)$ by an amount proportional to the covariance of the $n^{\text{th}}$ potential $f_n(\theta)$ with the residual error $\sum_{n=1}^N f_n(\theta) - \sum_{n=1}^N w_n f_n(\theta)$ under $\pi_w$. If required, it is not difficult to use the connection between derivatives of $\log Z(w)$ and moments of the sufficient statistic under $\pi_w$ to derive 2$^{\text{nd}}$ and higher order derivatives of $\mathrm{D}_{\mathrm{KL}}\left(\pi_w || \pi_1\right)$.

This provides a natural tool for optimizing the coreset construction objective in Eq. (5)—Monte Carlo estimates of sufficient statistic moments—and enables coreset construction without both the problematic selection of a Hilbert space (i.e., $\hat{\pi}$) and finite-dimensional projection error from past approaches. But obtaining Monte Carlo estimates requires sampling from $\pi_w$; the second key insight in this work is that as long as we build up the sparse approximation $w$ incrementally, the iterates will themselves be sparse. Therefore, using a standard Markov chain Monte Carlo algorithm [9] to obtain samples from $\pi_w$ for gradient estimation is actually not expensive—with cost $O(M)$ instead of $O(N)$—despite the potentially complicated form of $\pi_w$.

## 3.2 Greedy selection

One option to build up a coreset incrementally is to use a greedy approach (Algorithm 1) to select and subsequently reweight a single potential function at a time. For greedy selection, the naïve approach is to select the potential that provides the largest local decrease in KL divergence around the current weights $w$, i.e., selecting the potential with the largest covariance with the residual error per Eq. (9). However, since the weight $w_{n^\star}$ will then be optimized over $[0, \infty)$, the selection of the next potential to add should be invariant to scaling each potential $f_n$ by any positive constant. Thus we propose the use of the correlation—rather than the covariance—between $f_n$ and the residual error $f^T(1-w)$ as the selection criterion:

$$n^\star = \underset{n \in [N]}{\arg\max} \left\{ \begin{array}{ll} \left|\mathrm{Corr}_w\left[f_n, f^T(1-w)\right]\right| & w_n > 0 \\ \mathrm{Corr}_w\left[f_n, f^T(1-w)\right] & w_n = 0 \end{array} \right. . \quad (10)$$

Although seemingly ad-hoc, this modification will be placed on a solid information-geometric theoretical foundation in Proposition 1 (see also Eq. (34) in Appendix A). Note that since we do not have access to the exact correlations, we must use Monte Carlo estimates via sampling from $\pi_w$ for greedy selection. Given $S$ samples $(\theta_s)_{s=1}^S \overset{\text{i.i.d.}}{\sim} \pi_w$, these are given by the $N$-dimensional vector

$$\widehat{\mathrm{Corr}} = \mathrm{diag}\left[\frac{1}{S}\sum_{s=1}^S \hat{g}_s \hat{g}_s^T\right]^{-\frac{1}{2}} \left(\frac{1}{S}\sum_{s=1}^S \hat{g}_s \hat{g}_s^T (1-w)\right) \quad \hat{g}_s := \begin{bmatrix} f_1(\theta_s) \\ \vdots \\ f_N(\theta_s) \end{bmatrix} - \frac{1}{S}\sum_{r=1}^S \begin{bmatrix} f_1(\theta_r) \\ \vdots \\ f_N(\theta_r) \end{bmatrix}, \quad (11)$$

where $\mathrm{diag}\,[\cdot]$ returns a diagonal matrix with the same diagonal entries as its argument. The details of using the correlation estimate (Eq. (11)) in the greedy selection rule (Eq. (10)) to add points to the coreset are shown in lines 4–9 of Algorithm 1. Note that this computation has cost $O(NS)$. If $N$ is large enough that computing the entire vectors $\hat{g}_s \in \mathbb{R}^N$ is cost-prohibitive, one may instead compute $\hat{g}_s$ in Eq. (11) only for indices in $\mathcal{I} \cup \mathcal{U}$—where $\mathcal{I} = \{n \in [N] : w_n > 0\}$ is the set of active indices, and $\mathcal{U}$ is a uniformly selected subsample of $U \in [N]$ indices—and perform greedy selection only within these indices.

### 3.3 Weight update

After selecting a new potential function $n^\star$, we add it to the active set of indices $\mathcal{I} \subseteq [N]$ and update the weights by optimizing

$$w^\star = \underset{v \in \mathbb{R}^N}{\arg\min} \, \mathrm{D}_{\mathrm{KL}}\left(\pi_v || \pi\right) \quad \text{s.t.} \quad v \geq 0, \quad (1 - 1_{\mathcal{I}})^T v = 0. \tag{12}$$

In particular, we run $T$ steps of generating $S$ samples $(\theta_s)_{s=1}^S \overset{\text{i.i.d.}}{\sim} \pi_w$, computing a Monte Carlo estimate $D$ of the gradient $\nabla_w \mathrm{D}_{\mathrm{KL}}\left(\pi_w || \pi_1\right)$ based on Eq. (9),

$$D := -\frac{1}{S} \sum_{s=1}^S \hat{g}_s \hat{g}_s^T (1 - w) \in \mathbb{R}^N \quad \hat{g}_s \text{ as in Eq. (11)}, \tag{13}$$

and taking a stochastic gradient step $w_n \leftarrow w_n - \gamma_t D_n$ at step $t \in [T]$ for each $n \in \mathcal{I}$, using a typical learning rate $\gamma_t \propto t^{-1}$. The details of the weight update step are shown in lines 10–15 of Algorithm 1. As in the greedy selection step, the cost of each gradient step is $O(NS)$, due to the $\hat{g}_s^T 1$ term in the gradient. If $N$ is large enough that this computation is cost-prohibitive, one can use $\hat{g}_s$ computed only for indices in $\mathcal{I} \cup \mathcal{U}$, where $\mathcal{U}$ is a uniformly selected subsample of $U \in [N]$ indices.

## 4 The information geometry of coreset construction

The perspective of coresets as a sparse exponential family also enables the use of information geometry to derive a unifying connection between the variational formulation and previous constructions. In particular, the family of coreset posteriors defines a Riemannian statistical manifold $\mathcal{M} = \{\pi_w\}_{w \in \mathbb{R}_{\geq 0}^N}$ with chart $\mathcal{M} \to \mathbb{R}_{\geq 0}^N$, endowed with the *Fisher information metric* $G$ [61, p. 33,34],

$$G(w) = \int \pi_w(\theta) \nabla_w \log \pi_w(\theta) \nabla_w \log \pi_w(\theta)^T \mathrm{d}\theta = \nabla_w^2 \log Z(w) = \mathrm{Cov}_w\left[f\right]. \tag{14}$$

For any differentiable curve $\gamma : [0, 1] \to \mathbb{R}_{\geq 0}^N$, the metric defines a notion of path length,

$$L(\gamma) = \int_0^1 \sqrt{\frac{\mathrm{d}\gamma(t)}{\mathrm{d}t}^T G(\gamma(t)) \frac{\mathrm{d}\gamma(t)}{\mathrm{d}t}} \, \mathrm{d}t, \tag{15}$$

and a constant-speed curve of minimal length between any two points $w, w' \in \mathbb{R}_{\geq 0}^N$ is referred to as a *geodesic* [61, Thm. 5.2]. The geodesics are the generalization of straight lines in Euclidean space to curved Riemannian manifolds, such as $\mathcal{M}$. Using this information-geometric view, Proposition 1 shows that both Hilbert coreset construction (Eq. (3)) and the proposed greedy sparse variational inference procedure (Algorithm 1) attempt to directionally align the $\hat{w} \to w$ and $\hat{w} \to 1$ geodesics on $\mathcal{M}$ for $\hat{w}, w, 1 \in \mathbb{R}_{\geq 0}^N$ (reference, coreset, and true posterior weights, respectively) as illustrated in Fig. 1. The key difference is that Hilbert coreset construction uses a fixed reference point $\hat{w}$—corresponding to $\hat{\pi}$ in Eq. (3)—and thus operates entirely in a single tangent space of $\mathcal{M}$, while the proposed greedy method uses $\hat{w} = w$ and thus improves its tangent space approximation as the algorithm iterates. For this reason, we refer to the method in Section 3 as a *Riemannian coreset construction* algorithm. In addition to this unification of coreset construction methods, the geometric perspective also provides the means to show that the Hilbert coresets objective bounds the symmetrized coreset KL divergence $\mathrm{D}_{\mathrm{KL}}\left(\pi_w || \pi\right) + \mathrm{D}_{\mathrm{KL}}\left(\pi || \pi_w\right)$ if the Riemannian metric does not vary too much, as shown in Proposition 2. Incidentally, Lemma 3 in Appendix A—which is used to prove Proposition 2—also provides a nonnegative unbiased estimate of the symmetrized coreset KL divergence, which may be used for performance monitoring in practice.

---

**Algorithm 1** Greedy sparse stochastic variational inference

1: **procedure** SPARSEVI($f$, $\pi_0$, $S$, $T$, $(\gamma_t)_{t=1}^{\infty}$, $M$)
2:     $w \leftarrow 0 \in \mathbb{R}^N$, $\mathcal{I} \leftarrow \emptyset$
3:     **for** $m = 1, \dots, M$ **do**
           ▷ Take $S$ samples from the current coreset posterior approximation $\pi_w$
4:         $(\theta_s)_{s=1}^S \overset{\text{i.i.d.}}{\sim} \pi_w \propto \exp(w^T f(\theta))\pi_0(\theta)$
           ▷ Compute the $N$-dimensional potential vector for each sample
5:         $\hat{f}_s \leftarrow f(\theta_s) \in \mathbb{R}^N$ for $s \in [S]$, and $\bar{f} \leftarrow \frac{1}{S}\sum_{s=1}^S \hat{f}_s$
6:         $\hat{g}_s \leftarrow \hat{f}_s - \bar{f}$ for $s \in [S]$
           ▷ Estimate correlations between the potentials and the residual error
7:         $\widehat{\text{Corr}} \leftarrow \text{diag}\left[\frac{1}{S}\sum_{s=1}^S \hat{g}_s\hat{g}_s^T\right]^{-\frac{1}{2}}\left(\frac{1}{S}\sum_{s=1}^S \hat{g}_s\hat{g}_s^T(1-w)\right) \in \mathbb{R}^N$
           ▷ Add the best next potential to the coreset
8:         $n^\star \leftarrow \arg\max_{n \in [N]} |\widehat{\text{Corr}}_n| \mathbb{1}\left[n \in \mathcal{I}\right] + \widehat{\text{Corr}}_n \mathbb{1}\left[n \notin \mathcal{I}\right]$
9:         $\mathcal{I} \leftarrow \mathcal{I} \cup \{n^\star\}$
           ▷ Update all the active weights in $\mathcal{I}$ via stochastic gradient descent on $\text{D}_{\text{KL}}\left(\pi_w||\pi\right)$
10:        **for** $t = 1, \dots, T$ **do**
           ▷ Use samples from $\pi_w$ to estimate the gradient
11:            $(\theta_s)_{s=1}^S \overset{\text{i.i.d.}}{\sim} \pi_w \propto \exp(w^T f(\theta))\pi_0(\theta)$
12:            $\hat{f}_s \leftarrow f(\theta_s) \in \mathbb{R}^N$ for $s \in [S]$, and $\bar{f} \leftarrow \frac{1}{S}\sum_{s=1}^S \hat{f}_s$
13:            $\hat{g}_s \leftarrow \hat{f}_s - \bar{f}$ for $s \in [S]$
14:            $D \leftarrow -\frac{1}{S}\sum_{s=1}^S \hat{g}_s\hat{g}_s^T(1-w)$
           ▷ Take a stochastic gradient step for active indices in $\mathcal{I}$
15:            $w \leftarrow w - \gamma_t I_{\mathcal{I}}D$ where $I_{\mathcal{I}} := \sum_{n\in\mathcal{I}} 1_n 1_n^T$ is the diagonal indicator matrix for $\mathcal{I}$
16:        **end for**
17:    **end for**
18:    **return** $w$
19: **end procedure**

---

**Proposition 1.** *Suppose $\hat{\pi}$ in Eq. (3) satisfies $\hat{\pi} = \pi_{\hat{w}}$ for a set of weights $\hat{w} \in \mathbb{R}_{\geq 0}^N$. For $u, v \in \mathbb{R}_{\geq 0}^N$, let $\xi_{u \to v}$ denote the initial tangent of the $u \to v$ geodesic on $\mathcal{M}$, and $\langle \cdot, \cdot \rangle_u$ denote the inner product under the Riemannian metric $G(u)$ with induced norm $\|\cdot\|_u$. Then Hilbert coreset construction in Eq. (3) is equivalent to*

$$w^\star = \underset{w \in \mathbb{R}^N}{\arg\min} \ \|\xi_{\hat{w} \to 1} - \xi_{\hat{w} \to w}\|_{\hat{w}} \quad s.t. \quad w \geq 0, \ \|w\|_0 \leq M, \tag{16}$$

*and each greedy selection step of Riemannian coreset construction in Eq. (10) is equivalent to*

$$n^\star = \underset{n \in [N], t_n \in \mathbb{R}}{\arg\min} \ \|\xi_{w \to 1} - \xi_{w \to w + t_n 1_n}\|_w \quad s.t. \quad \forall n \notin \mathcal{I}, \ t_n > 0. \tag{17}$$

**Proposition 2.** *Suppose $\hat{\pi}$ in Eq. (3) satisfies $\hat{\pi} = \pi_{\hat{w}}$ for a set of weights $\hat{w} \in \mathbb{R}_{\geq 0}^N$. Then if $J_{\hat{\pi}}(w)$ is the objective function in Eq. (3),*

$$\text{D}_{\text{KL}}\left(\pi||\pi_w\right) + \text{D}_{\text{KL}}\left(\pi_w||\pi\right) \leq C_{\hat{\pi}}(w) \cdot J_{\hat{\pi}}(w), \tag{18}$$

*where $C_{\hat{\pi}}(w) := \mathbb{E}_{U \sim \text{Unif}[0,1]}\left[\lambda_{\max}\left(G(\hat{w})^{-1/2}G((1-U)w + U1)G(\hat{w})^{-1/2}\right)\right]$. In particular, if $\nabla_w^2 \log Z(w)$ is constant in $w \in \mathbb{R}_{\geq 0}^N$, then $C_{\hat{\pi}}(w) = 1$.*

## 5  Experiments

In this section, we compare the quality of coresets constructed via the proposed `SparseVI` greedy coreset construction method, uniform random subsampling, and Hilbert coreset construction (GIGA [32]). In particular, for GIGA we used a 100-dimensional random projection generated from a Gaussian $\hat{\pi}$ with two parametrizations: one with mean and covariance set using the moments of the exact posterior (Optimal) which is a benchmark but is not possible to achieve in practice; and one

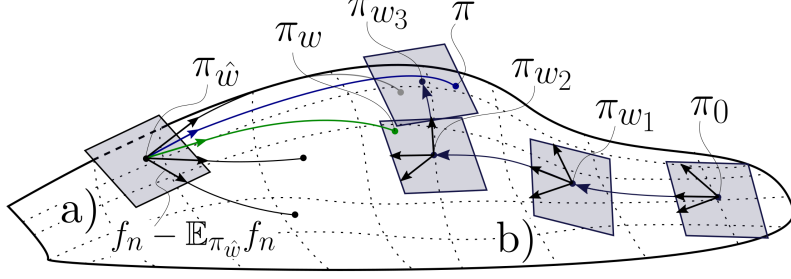

Figure 1: Information-geometric view of greedy coreset construction on the coreset manifold $\mathcal{M}$. (1a): Hilbert coreset construction, with weighting distribution $\pi_{\hat{w}}$, full posterior $\pi$, coreset posterior $\pi_w$, and arrows denoting initial geodesic directions from $\hat{w}$ towards new datapoints. (1b): Riemannian coreset construction, with the path of posterior approximations $\pi_{w_t}$, $t = 0, \ldots, 3$, and arrows denoting initial geodesic directions towards new datapoints to add within each tangent plane.

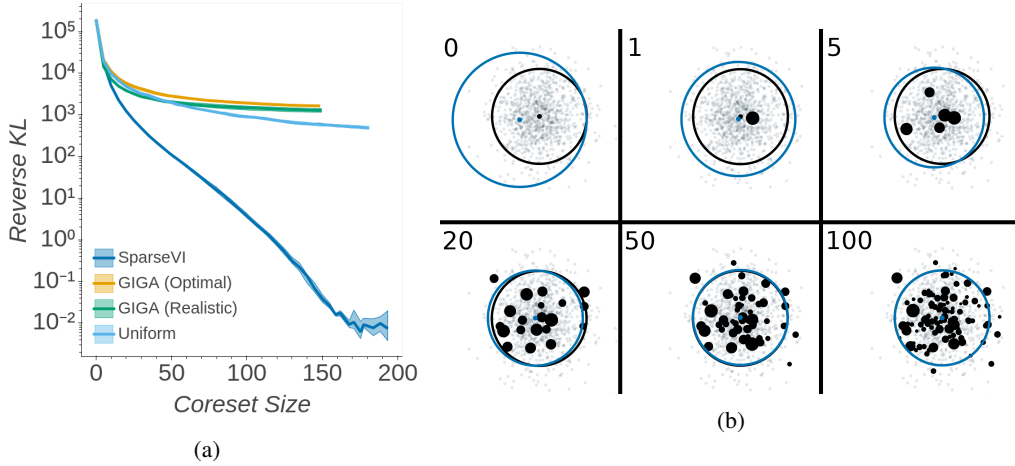

Figure 2: (2a): Synthetic comparison of coreset construction methods. Solid lines show the median KL divergence over 10 trials, with 25[th] and 75[th] percentiles shown by shaded areas. (2b): 2D projection of coresets after 0, 1, 5, 20, 50, and 100 iterations via `SparseVI`. True/coreset posterior and $2\sigma$-predictive ellipses are shown in black/blue respectively. Coreset points are black with radius denoting weight.

with mean and covariance uniformly distributed between the prior and the posterior with 75% relative noise added (Realistic) to simulate the choice of $\hat{\pi}$ without exact posterior information. Experiments were performed on a machine with an Intel i7 8700K processor and 32GB memory; code is available at `www.github.com/trevorcampbell/bayesian-coresets`.

## 5.1 Synthetic Gaussian posterior inference

We first compared the coreset construction algorithms on a synthetic example involving posterior inference for the mean of a $d$-dimensional Gaussian with Gaussian observations,

$$\theta \sim \mathcal{N}(\mu_0, \Sigma_0) \qquad x_n \overset{\text{i.i.d.}}{\sim} \mathcal{N}(\theta, \Sigma), \quad n = 1, \ldots, N. \tag{19}$$

We selected this example because it decouples the evaluation of the coreset construction methods from the concerns of stochastic optimization and approximate posterior inference: the coreset posterior $\pi_w$ is a Gaussian $\pi_w = \mathcal{N}(\mu_w, \Sigma_w)$ with closed-form expressions for the parameters as well as covariance (see Appendix B for the derivation),

$$\Sigma_w = \left(\Sigma_0^{-1} + \sum_{n=1}^{N} w_n \Sigma^{-1}\right)^{-1} \qquad \mu_w = \Sigma_w \left(\Sigma_0^{-1} \mu_0 + \Sigma^{-1} \sum_{n=1}^{N} w_n x_n\right) \tag{20}$$

$$\text{Cov}_w [f_n, f_m] = {}^1\!/_2 \operatorname{tr} \Psi^T \Psi + \nu_m^T \Psi \nu_n, \tag{21}$$

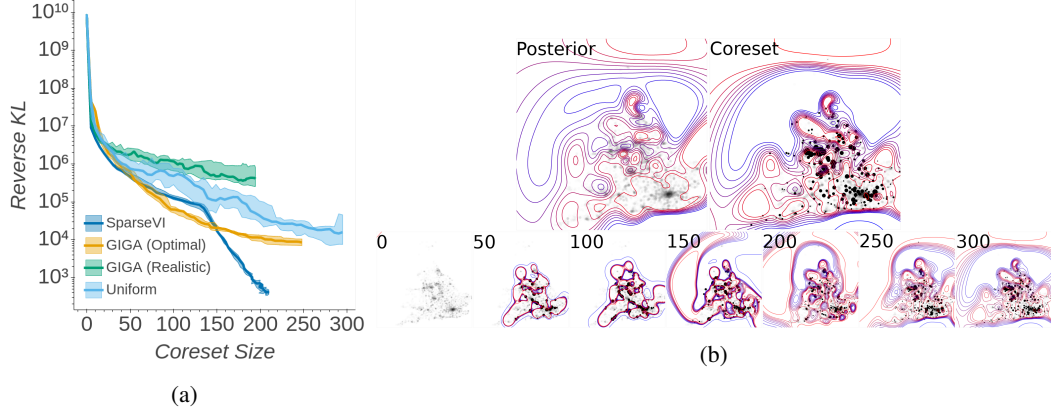

(a)

(b)

Figure 3: (3a): Coreset construction on regression of housing prices using radial basis functions in the UK Land Registry data. Solid lines show the median KL divergence over 10 trials, with $25^{\text{th}}$ and $75^{\text{th}}$ percentiles shown by shaded areas. (3b): Posterior mean contours with coresets of size 0–300 via `SparseVI` compared with the exact posterior. Posterior and final coreset highlighted in the top row. Coreset points are black with radius denoting weight.

where $\Sigma = QQ^T$, $\nu_n := Q^{-1}(x_n - \mu_w)$ and $\Psi := Q^{-1}\Sigma_w Q^{-T}$. Thus the greedy selection and weight update can be performed without Monte Carlo estimation. We set $\Sigma_0 = \Sigma = I$, $\mu_0 = 0$, $d = 200$, and $N = 1,000$. We used a learning rate of $\gamma_t = t^{-1}$, $T = 100$ weight update optimization iterations, and $M = 200$ greedy iterations, although note that this is an upper bound on the size of the coreset as the same data point may be selected multiple times. The results in Fig. 2 demonstrate that the use of a fixed weighting function $\hat{\pi}$ (and thus, a fixed tangent plane on the coreset manifold) fundamentally limits the quality coresets via past algorithms. In contrast, the proposed greedy algorithm is "manifold-aware" and is able to continually improve the approximation, resulting in orders-of-magnitude improvements in KL divergence to the true posterior.

## 5.2 Bayesian radial basis function regression

Next, we compared the coreset construction algorithms on Bayesian basis function regression for $N = 10,000$ records of house sale log-price $y_n \in \mathbb{R}$ as a function of latitude / longitude coordinates $x_n \in \mathbb{R}^2$ in the UK.[3] The regression problem involved inference for the coefficients $\alpha \in \mathbb{R}^K$ in a linear combination of radial basis functions $b_k(x) = \exp(-1/2\sigma_k^2(x - \mu_k)^2)$, $k = 1, \dots, K$,

$$y_n = b_n^T \alpha + \epsilon_n \quad \epsilon_n \overset{\text{i.i.d.}}{\sim} \mathcal{N}(0, \sigma^2) \quad b_n = [\ b_1(x_n) \quad \cdots \quad b_K(x_n)\ ]^T \quad \alpha \sim \mathcal{N}(\mu_0, \sigma_0^2 I). \quad (22)$$

We generated 50 basis functions for each of 6 scales $\sigma_k \in \{0.2, 0.4, 0.8, 1.2, 1.6, 2.0\}$ by generating means $\mu_k$ uniformly from the data, and added one additional near-constant basis with scale 100 and mean corresponding to the mean latitude and longitude of the data. This resulted in $K = 301$ total basis functions and thus a 301-dimensional regression problem. We set the prior and noise parameters $\mu_0, \sigma_0^2, \sigma^2$ equal to the empirical mean, second moment, and variance of the price paid $(y_n)_{n=1}^N$ across the whole dataset, respectively. As in Section 5.1, the posterior and log-likelihood covariances are available in closed form, and all algorithmic steps can be performed without Monte Carlo. In particular, $\pi_w = \mathcal{N}(\mu_w, \Sigma_w)$, where (see Appendix B for the derivation)

$$\Sigma_w = (\Sigma_0^{-1} + \sigma^{-2}\sum_{n=1}^N w_n b_n b_n^T)^{-1} \quad \text{and} \quad \mu_w = \Sigma_w(\Sigma_0^{-1}\mu_0 + \sigma^{-2}\sum_{n=1}^N w_n y_n b_n) \quad (23)$$

$$\text{Cov}_w[f_n, f_m] = \sigma^{-4}\left(\nu_n \nu_m \beta_n^T \beta_m + 1/2(\beta_n^T \beta_m)^2\right). \quad (24)$$

where $\nu_n := y_n - \mu_w^T b_n$, $\Sigma_w = LL^T$, and $\beta_n := L^T b_n$. We used a learning rate of $\gamma_t = t^{-1}$, $T = 100$ optimization steps, and $M = 300$ greedy iterations, although again note that this is an upper

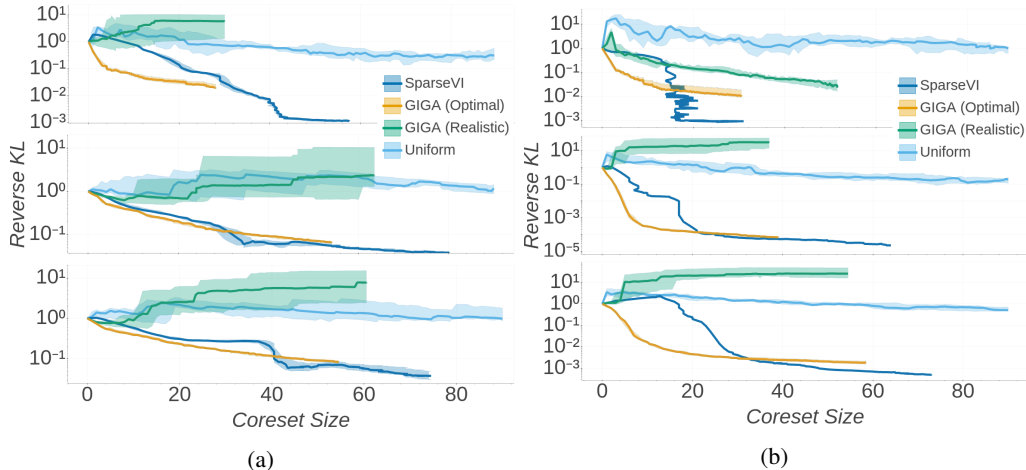

Figure 4: The results of the logistic (4a) and Poisson (4b) regression experiments. Plots show the median KL divergence (estimated using the Laplace approximation [62] and normalized by the value for the prior) across 10 trials, with $25^{th}$ and $75^{th}$ percentiles shown by shaded areas. From top to bottom, (4a) shows the results for logistic regression on synthetic, chemical reactivities, and phishing websites data, while (4b) shows the results for Poisson regression on synthetic, bike trips, and airport delays data. See Appendix C for details.

bound on the coreset size. The results in Fig. 3 generally align with those from the previous synthetic experiment. The proposed sparse variational inference formulation builds coresets of comparable quality to Hilbert coreset construction (when given the exact posterior for $\hat{\pi}$) up to a size of about 150. Beyond this point, past methods become limited by their fixed tangent plane approximation while the proposed method continues to improve. This experiment also highlights the sensitivity of past methods to the choice of $\hat{\pi}$: uniform subsampling outperforms GIGA with a realistic choice of $\hat{\pi}$.

### 5.3 Bayesian logistic and Poisson regression

Finally, we compared the methods on logistic and Poisson regression applied to six datasets (details may be found in Appendix C) with $N = 500$ and dimension ranging from 2-15. We used $M = 100$ greedy iterations, $S = 100$ samples for Monte Carlo covariance estimation, and $T = 500$ optimization iterations with learning rate $\gamma_t = 0.5t^{-1}$. Fig. 4 shows the result of this test, demonstrating that the proposed greedy sparse VI method successfully recovers a coreset with divergence from the exact posterior as low or lower than GIGA with without having the benefit of a user-specified weighting function. Note that there is a computational price to pay for this level of automation; Fig. 5, Appendix C shows that `SparseVI` is significantly slower than Hilbert coreset construction via GIGA [32], primarily due to the expensive gradient descent weight update. However, if we remove `GIGA (Optimal)` from consideration due to its unrealistic use of $\hat{\pi} \approx \pi_1$, `SparseVI` is the only practical coreset construction algorithm that reduces the KL divergence to the posterior appreciably for reasonable coreset sizes. We leave improvements to computational cost for future work.

## 6 Conclusion

This paper introduced sparse variational inference for Bayesian coreset construction. By exploiting the fact that coreset posteriors form an exponential family, a greedy algorithm as well as a unifying Riemannian information-geometric view of present and past coreset constructions were developed. Future work includes extending sparse VI to improved optimization techniques beyond greedy methods, and reducing computational cost.

**Acknowledgments** T. Campbell and B. Beronov are supported by National Sciences and Engineering Research Council of Canada (NSERC) Discovery Grants. T. Campbell is additionally supported by an NSERC Discovery Launch Supplement.

## Footnotes

[1]Throughout, $[N] := \{1, \ldots, N\}$, $1$ and $0$ are the constant vectors of all 1s / 0s respectively (the dimension will be clear from context), $1_{\mathcal{A}}$ is the indicator vector for $\mathcal{A} \subseteq [N]$, and $1_n$ is the indicator vector for $n \in [N]$.

[2]In [31], the $\mathbb{E}_{\hat{\pi}}\left[f_n\right]$ term was missing; it is necessary to account for the shift-invariance of potentials.

[3]This dataset was constructed by merging housing prices from the UK land registry data `https://www.gov.uk/government/statistical-data-sets/price-paid-data-downloads` with latitude & longitude coordinates from the Geonames postal code data `http://download.geonames.org/export/zip/`.

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
