[Supplementary Material]

# A Proofs of results in the Riemannian information geometry section

*Proof of Proposition 1.* By Eq. (14), we have that $G(\hat{w}) = \text{Cov}_{\hat{w}}[f]$, and so

$$\mathbb{E}_{\hat{\pi}}\left[\left(\sum_{n=1}^{N}(1-w_n)g_n\right)^2\right] = \mathbb{E}_{\hat{\pi}}\left[\left((1-w)^T(f-\mathbb{E}_{\hat{\pi}}[f])\right)^2\right] \tag{25}$$

$$= (1-w)^T \mathbb{E}_{\hat{\pi}}\left[(f-\mathbb{E}_{\hat{\pi}}[f])(f-\mathbb{E}_{\hat{\pi}}[f])^T\right](1-w) \tag{26}$$

$$= (1-w)^T G(\hat{w})(1-w) \tag{27}$$

$$= ((1-\hat{w}) - (w-\hat{w}))^T G(\hat{w})((1-\hat{w}) - (w-\hat{w})) \tag{28}$$

$$= (\xi_{\hat{w}\to 1} - \xi_{\hat{w}\to w})^T G(\hat{w})(\xi_{\hat{w}\to 1} - \xi_{\hat{w}\to w}), \tag{29}$$

yielding the first result. Next, note that

$$\|\xi_{w\to 1} - \xi_{w\to w+t_n 1_n}\|_w = t_n^2 1_n^T G(w) 1_n - 2t_n 1_n^T G(w)(1-w) + (1-w)^T G(w)(1-w), \tag{30}$$

and minimizing over $t_n$ yields

$$t_n^\star = \frac{1_n^T G(w)(1-w)}{1_n^T G(w) 1_n} \quad \text{or} \quad t_n^\star = \max\left\{0, \frac{1_n^T G(w)(1-w)}{1_n^T G(w) 1_n}\right\} \tag{31}$$

if $t_n$ is unconstrained or positive-constrained, respectively. Substituting back into the norm and using the definition of norms and inner products via the Riemannian metric $G(w)$,

$$\|\ldots\|_w = \begin{cases} \|1-w\|_w^2 \left(1 - \left(\left\langle \frac{1_n}{\|1_n\|_w}, \frac{1-w}{\|1-w\|_w}\right\rangle_w\right)^2\right) & t_n \in \mathbb{R} \\ \|1-w\|_w^2 \left(1 - \left(\max\left\{0, \left\langle \frac{1_n}{\|1_n\|_w}, \frac{1-w}{\|1-w\|_w}\right\rangle_w\right\}\right)^2\right) & t_n > 0 \end{cases} \tag{32}$$

Finally, expressing the inner product explicitly,

$$\left\langle \frac{1_n}{\|1_n\|_w}, \frac{1-w}{\|1-w\|_w}\right\rangle_w = \frac{\mathbb{E}_w\left[(f_n - \mathbb{E}_w f_n)(f - \mathbb{E}_w f)^T(1-w)\right]}{\sqrt{\mathbb{E}_w\left[(f_n - \mathbb{E}_w f_n)^2\right]\mathbb{E}_w\left[\left((f-\mathbb{E}_w f)^T(1-w)\right)^2\right]}} \tag{33}$$

$$= \text{Corr}_w\left[f_n, (1-w)^T f\right], \tag{34}$$

yielding the second result. $\square$

**Lemma 3.** *Define the path $\gamma(t) = (1-t)w + t1$. Then*

$$\text{D}_{\text{KL}}(\pi_w || \pi) = 2(1-w)^T \mathbb{E}\left[G(\gamma(T))\right](1-w) \quad T \sim \text{Beta}(1,2) \tag{35}$$

$$\text{D}_{\text{KL}}(\pi || \pi_w) = 2(1-w)^T \mathbb{E}\left[G(\gamma(S))\right](1-w) \quad S \sim \text{Beta}(2,1) \tag{36}$$

$$\text{D}_{\text{KL}}(\pi_w || \pi) + \text{D}_{\text{KL}}(\pi || \pi_w) = (1-w)^T \mathbb{E}\left[G(\gamma(U))\right](1-w) \quad U \sim \text{Unif}[0,1]. \tag{37}$$

*Proof.* Here we use prime notation for univariate differentiation. For any twice differentiable function $h : [0,1] \to \mathbb{R}$, the Taylor remainder theorem states that

$$h(1) = h(0) + h'(0) + \int_0^1 h''(t)(1-t)\text{d}t. \tag{38}$$

Let $\gamma : [0,1] \to \mathbb{R}_{\geq 0}^N$ be any twice-differentiable path satisfying $\gamma(0) = w$, $\gamma(1) = 1$, $\gamma'(0) = \gamma'(1) = 1 - w$. Then if the log partition $\log Z(w)$ is also twice differentiable, setting $h(t) = \log Z(\gamma(t))$ shows that

$$\log Z(1) = \log Z(w) + (1-w)^T \nabla \log Z(w) +$$
$$\int_0^1 (1-t)\left(\gamma'(t)^T \nabla^2 \log Z(\gamma(t))\gamma'(t) + \gamma''(t)^T \nabla \log Z(\gamma(t))\right)\text{d}t. \tag{39}$$

Substituting into Eq. (8) yields

$$\text{D}_{\text{KL}}(\pi_w || \pi) = \int_0^1 (1-t)\left(\gamma'(t)^T \nabla^2 \log Z(\gamma(t))\gamma'(t) + \gamma''(t)^T \nabla \log Z(\gamma(t))\right)\text{d}t. \tag{40}$$

The same logic follows with $\mathrm{D}_{\mathrm{KL}}\left(\pi||\pi_w\right)$, using a path $\zeta$ from 1 to $w$ with $\zeta'(0) = \zeta'(1) = w - 1$. So selecting the path $\zeta(t) = \gamma(1-t)$ and using the transformation of variables $t \to 1 - s$,

$$\mathrm{D}_{\mathrm{KL}}\left(\pi||\pi_w\right) = \int_0^1 t\left(\gamma'(t)^T \nabla^2 \log Z(\gamma(t))\gamma'(t) + \gamma''(t)^T \nabla \log Z(\gamma(t))\right) \mathrm{d}t. \tag{41}$$

Adding the two expressions together makes the $t$ and $1-t$ terms cancel, and noting that the densities $\propto t$ and $\propto 1 - t$ are beta densities yields the stated result. $\qquad\square$

*Proof of Proposition 2.* By Lemma 3 we have that

$$\mathrm{D}_{\mathrm{KL}}\left(\pi||\pi_w\right) + \mathrm{D}_{\mathrm{KL}}\left(\pi_w||\pi\right) \le (1-w)^T \int_0^1 G(\gamma(t))\mathrm{d}t(1-w). \tag{42}$$

Multiplying and dividing by $J_{\hat\pi}(w) = (1-w)^T\nabla^2 \log Z(\hat w)(1-w)$ from Eq. (29), defining $v := \frac{\nabla^2 \log Z(\hat w)^{1/2}(1-w)}{\left\|\nabla^2 \log Z(\hat w)^{1/2}(1-w)\right\|}$, and defining $\tilde{G}(t) = G(\hat w)^{-1/2}G(\gamma(t))G(\hat w)^{-1/2}$ yields

$$\mathrm{D}_{\mathrm{KL}}\left(\pi_w||\pi\right) = J_{\hat\pi}(w)\left(\int_0^1 (1-t)v^T\tilde{G}(t)v\mathrm{d}t\right) \le J_{\hat\pi}(w)\left(\int_0^1 (1-t)\lambda_{\max}\left(\tilde{G}(t)\right)\mathrm{d}t\right). \tag{43}$$

Likewise,

$$\mathrm{D}_{\mathrm{KL}}\left(\pi||\pi_w\right) \le J_{\hat\pi}(w)\left(\int_0^1 tv^T\tilde{G}(t)v\mathrm{d}t\right) \le J_{\hat\pi}(w)\left(\int_0^1 t\lambda_{\max}\left(\tilde{G}(t)\right)\mathrm{d}t\right). \tag{44}$$

Adding these equations yields the stated result. $\qquad\square$

# B Weighted posterior and sufficient statistic covariance derivations

## B.1 Simple Gaussian inference

The log likelihood for datapoint $x_n$ is (dropping normalization constants)

$$f_n(\theta) = -\frac{1}{2}\left(x_n - \theta\right)^T \Sigma^{-1}\left(x_n - \theta\right), \tag{45}$$

so the $w$-weighted log-posterior is (again, up to normalization constants)

$$\theta^T\left(\Sigma_0^{-1}\mu_0 + \Sigma^{-1}\sum_{n=1}^N w_n x_n\right) - \frac{1}{2}\theta^T\left(\Sigma_0^{-1} + \sum_{n=1}^N w_n\Sigma^{-1}\right)\theta. \tag{46}$$

Completing the square yields Eq. (20). The first moment of the log-likelihood under the coreset posterior $\theta \sim \mathcal{N}(\mu_w, \Sigma_w)$ is:

$$\mathbb{E}_w\left[f_n(\theta)\right] = -\frac{1}{2}\mathbb{E}_w\left[(x_n - \theta)^T \Sigma^{-1}(x_n - \theta)\right] \tag{47}$$

$$= -\frac{1}{2}\operatorname{tr}\Sigma^{-1}\Sigma_w - \frac{1}{2}(\mu_w - x_n)^T \Sigma^{-1}(\mu_w - x_n) \tag{48}$$

$$= -\frac{1}{2}\operatorname{tr}\Psi - \frac{1}{2}\|\nu_n\|^2, \tag{49}$$

where $\Psi = Q^{-1}\Sigma_w Q^{-T}$, $\nu_n = Q^{-1}(x_n - \mu_w)$, and $Q$ is the Cholesky decomposition of $\Sigma$, i.e., $\Sigma = QQ^T$. Defining $z \sim \mathcal{N}(0, \Psi)$, its second moment is

$$\mathbb{E}_w\left[f_n(\theta)f_m(\theta)\right] = \frac{1}{4}\mathbb{E}_w\left[(x_n - \theta)^T \Sigma^{-1}(x_n - \theta)(x_m - \theta)^T \Sigma^{-1}(x_m - \theta)\right] \tag{50}$$

$$= \frac{1}{4}\mathbb{E}_w\left[(z - \nu_n)^T(z - \nu_n)(z - \nu_m)^T(z - \nu_m)\right] \tag{51}$$

and by expanding and ignoring odd-order terms (which have 0 expectation),

$$= \frac{1}{4}\mathbb{E}_w\left[z^Tzz^Tz + z^Tz\nu_m^T\nu_m + 4z^T\nu_n z^T\nu_m + \nu_n^T\nu_n z^Tz + \nu_n^T\nu_n\nu_m^T\nu_m\right] \tag{52}$$

$$= \frac{1}{4}\left((\operatorname{tr}\Psi)^2 + 2\operatorname{tr}\Psi^T\Psi + \|\nu_m\|^2\|\nu_n\|^2 + \left(\|\nu_m\|^2 + \|\nu_n\|^2\right)\operatorname{tr}\Psi + 4\nu_m^T\Psi\nu_n\right). \tag{53}$$

So therefore,

$$\operatorname{Cov}_w\left[f_n, f_m\right] = \mathbb{E}_w\left[f_n(\theta)f_m(\theta)\right] - \mathbb{E}_w\left[f_n(\theta)\right]\mathbb{E}_w\left[f_m(\theta)\right] \tag{54}$$

$$= \nu_m^T\Psi\nu_n + \frac{1}{2}\operatorname{tr}\Psi^T\Psi. \tag{55}$$

## B.2 Bayesian radial basis regression

The log likelihood for datapoint $n$ is (dropping normalization constants)

$$f_n(\alpha) = -\frac{1}{2\sigma^2}\left(y_n - \alpha^T b_n\right)^2, \tag{56}$$

so the $w$-weighted log-posterior is (again, up to normalization constants)

$$\alpha^T \left(\sigma_0^{-2}\mu_0 + \sigma^{-2}\sum_{n=1}^{N} w_n y_n b_n\right) - \frac{1}{2}\alpha^T \left(\sigma_0^{-2} I + \sigma^{-2}\sum_{n=1}^{N} w_n b_n b_n^T\right)\alpha. \tag{57}$$

Completing the square yields Eq. (24). The first moment of the log-likelihood under the coreset posterior $\alpha \sim \mathcal{N}(\mu_w, \Sigma_w)$ is:

$$\mathbb{E}_w\left[f_n(\alpha)\right] = -\frac{1}{2\sigma^2}\mathbb{E}\left[\left(y_n - \mu_w^T b_n\right)^2 + (\mu_w - \alpha)^T b_n b_n^T (\mu_w - \alpha)\right] \tag{58}$$

$$= -\frac{1}{2\sigma^2}\left(\nu_n^2 + \operatorname{tr} b_n b_n^T \mathbb{E}\left[(\mu_w - \alpha)(\mu_w - \alpha)^T\right]\right) \tag{59}$$

$$= -\frac{1}{2\sigma^2}\left(\nu_n^2 + \|\beta_n\|^2\right). \tag{60}$$

where $\nu_n := (y_n - \mu_w^T b_n)$, $\Sigma_w = LL^T$, and $\beta_n = L^T b_n$. Defining $Z = L^{-1}(\alpha - \mu_w) \sim \mathcal{N}(0, I)$, the second moment is

$$\mathbb{E}_w\left[f_n(\alpha)f_m(\alpha)\right] = \frac{1}{4\sigma^4}\mathbb{E}_w\left[\left(y_n - \alpha^T b_n\right)^2\left(y_m - \alpha^T b_m\right)^2\right] \tag{61}$$

$$= \frac{1}{4\sigma^4}\mathbb{E}_w\left[\left(\nu_n - Z^T\beta_n\right)^2\left(\nu_m - Z^T\beta_m\right)^2\right]. \tag{62}$$

Expanding and ignoring odd-order terms which have expectation 0,

$$= \frac{1}{4\sigma^4}\left(\nu_n^2\nu_m^2 + \nu_n^2\|\beta_m\|^2 + 4\nu_n\nu_m\beta_n^T\beta_m + \nu_m^2\|\beta_n\|^2 + \sum_{i,j}\beta_{ni}^2\beta_{mj}^2 + 2\beta_{ni}\beta_{mi}\beta_{nj}\beta_{mj}\right) \tag{63}$$

$$= \frac{1}{4\sigma^4}\left(\nu_n^2\nu_m^2 + \nu_n^2\|\beta_m\|^2 + 4\nu_n\nu_m\beta_n^T\beta_m + \nu_m^2\|\beta_n\|^2 + \|\beta_n\|^2\|\beta_m\|^2 + 2(\beta_n^T\beta_m)^2\right). \tag{64}$$

Therefore, the covariance is

$$\operatorname{Cov}_w\left[f_n, f_m\right] = \mathbb{E}_w\left[f_n(\alpha)f_m(\alpha)\right] - \mathbb{E}_w\left[f_n(\alpha)\right]\mathbb{E}_w\left[f_m(\alpha)\right] \tag{65}$$

$$= \frac{1}{\sigma^4}\left(\nu_n\nu_m\beta_n^T\beta_m + \frac{1}{2}(\beta_n^T\beta_m)^2\right). \tag{66}$$

# C   Details of the Logistic / Poisson regression experiment

In logistic regression, we are given a set of data points $(x_n, y_n)_{n=1}^{N}$ each consisting of a feature $x_n \in \mathbb{R}^D$ and a label $y_n \in \{-1, 1\}$. The goal is to infer the posterior distribution of the latent parameter $\theta \in \mathbb{R}^{D+1}$ in the following model:

$$y_n \mid x_n, \theta \overset{\text{indep}}{\sim} \operatorname{Bern}\left(\frac{1}{1 + e^{-z_n^T\theta}}\right) \qquad z_n := \begin{bmatrix} x_n \\ 1 \end{bmatrix}. \tag{67}$$

We used three datasets (each subsampled to $N = 500$ data points) in the logistic regression experiment: a synthetic dataset with covariate $x_n \in \mathbb{R}^2$ sampled i.i.d. from $\mathcal{N}(0, I)$, and label $y_n \in \{-1, 1\}$ generated from the logistic likelihood with parameter $\theta = [3, 3, 0]^T$; a phishing websites dataset reduced to $D = 10$ features via principal component analysis; and a chemical reactivity dataset with $D = 10$ features. The original phishing and chemical reactivities datasets are available online at `https://www.csie.ntu.edu.tw/~cjlin/libsvmtools/datasets/binary.html` and `http://komarix.org/ac/ds/`. Preprocessed versions for the experiments in this paper are available at `https://www.github.com/trevorcampbell/bayesian-coresets/`.

In Poisson regression, we are given a set of data points $(x_n, y_n)_{n=1}^{N}$, each consisting of a feature $x_n \in \mathbb{R}^D$ and a count $y_n \in \mathbb{N}$. The goal is to infer the posterior distribution of the latent parameter $\theta \in \mathbb{R}^{D+1}$ in the following model:

$$y_n \mid x_n, \theta \overset{\text{indep}}{\sim} \operatorname{Poiss}\left(\log\left(1 + e^{z_n^T\theta}\right)\right) \qquad z_n := \begin{bmatrix} x_n \\ 1 \end{bmatrix}. \tag{68}$$

Figure 5: Computation times for the logistic (5a) and Poisson (5b) regression experiments. Plots show the median KL divergence (estimated using the Laplace approximation [62] and normalized by the value for the prior) across 10 trials, with $25^{\text{th}}$ and $75^{\text{th}}$ percentiles shown by shaded areas. From top to bottom, (5a) shows the results for logistic regression on synthetic, chemical reactivities, and phishing websites data, while (5b) shows the results for Poisson regression on synthetic, bike trips, and airport delays data.

We used three additional datasets (each subsampled to $N = 500$ data points) in the Poisson regression experiment: a synthetic dataset with covariate $x_n \in \mathbb{R}$ sampled i.i.d. from $\mathcal{N}(0, 1)$, and count $y_n \in \mathbb{N}$ generated from the Poisson likelihood with $\theta = [1, 0]^T$; a bikeshare dataset with $D = 8$ features, relating the weather and seasonal information to the number of bike trips taken in an urban area; and an airport delays dataset with $D = 15$ features, relating daily weather information to the number of flights leaving an airport with a delay of more than 15 minutes. The original bikeshare dataset is available online at `http://archive.ics.uci.edu/ml/datasets/Bike+Sharing+Dataset`, and the airport delays dataset was constructed using flight delay data from `http://stat-computing.org/dataexpo/2009/the-data.html` and historical weather information from `https://www.wunderground.com/history/`. Preprocessed versions for the experiments in this paper are available at `https://www.github.com/trevorcampbell/bayesian-coresets/`.

## D   SparseVI optimization alternatives

In the main text, we proposed one particular instantiation of sparse variational inference based on a greedy iterative method and full gradient-descent-based weight update. There are many possible variations on this theme; we highlight a few potential directions to explore in future work below.

### D.1   Single weight update

Rather than updating all the active weights, one might scale the current weights $w$ while adding the new component $1_{n^\star}$ via

$$w^\star = \omega(\alpha^\star, \beta^\star) \quad \alpha^\star, \beta^\star = \underset{\alpha,\beta \geq 0}{\arg\min} \, \mathrm{D_{KL}}\left(\pi_{\omega(\alpha,\beta)} || \pi\right) \quad \text{s.t.} \quad \alpha, \beta \geq 0, \tag{69}$$

where $\omega(\alpha, \beta) := \beta w + \alpha 1_{n^\star}$. To optimize, one would use Monte Carlo estimates of the gradients

$$\begin{bmatrix} \frac{\partial}{\partial \beta} \\ \frac{\partial}{\partial \alpha} \end{bmatrix} \mathrm{D_{KL}}\left(\pi_{\omega(\alpha,\beta)} || \pi\right) = \begin{bmatrix} w & 1_{n^\star} \end{bmatrix}^T \nabla_w \, \mathrm{D_{KL}}\left(\pi_w || \pi\right)|_{w=\omega(\alpha,\beta)}. \tag{70}$$

### D.2   Quadratic weight update

The major computational cost in `SparseVI` is the weight updates in Section 3.3: for each gradient step, one must simulate a set of samples from $\pi_w$, compute all of the potentials, and finally compute the Monte Carlo gradient estimate. Rather than optimizing the weights exactly, one might minimize a quadratic expansion of the KL divergence at the point $w$,

$$\mathrm{D_{KL}}\left(\pi_v || \pi\right) \approx \mathrm{D_{KL}}\left(\pi_w || \pi\right) + (v-w)^T \nabla_w \mathrm{D_{KL}}\left(\pi_w || \pi\right) + \frac{1}{2}(v-w)^T \nabla_w^2 \mathrm{D_{KL}}\left(\pi_w || \pi\right)(v-w), \tag{71}$$

with Monte Carlo estimates of the gradient $D$ and Hessian $H$ based on the potential vector approximations $(\hat{g}_s)_{s=1}^S$ already obtained in the greedy selection step,

$$D := -\frac{1}{S}\sum_{s=1}^S \hat{g}_s \hat{g}_s^T (1-w), \qquad LL^T = H := \frac{1}{S}\sum_{s=1}^S \hat{g}_s \hat{g}_s^T (1 - \hat{g}_s^T(1-w)). \qquad (72)$$

Since Eq. (71) is quadratic in $v$ or $\alpha, \beta$ (depending on which type of weight update is used), the resulting weight update optimization is a nonnegative least squares problem,

$$v^\star = \underset{v \in \mathbb{R}^N, v \geq 0}{\arg\min} \left\| L^T v - \left( L^T w - L^{-1}D \right) \right\|^2 \quad \text{s.t.} \quad \begin{cases} (1-1_{\mathcal{I}})^T v = 0 & \text{(fully corrective)} \\ v = \omega(\alpha, \beta) & \text{(single-update)} \end{cases}. \qquad (73)$$

Upon solving the problem for $v^\star$, update the weights via $w \leftarrow (1-\gamma_t)w + \gamma_t v^\star$ with a learning schedule $\gamma_t \geq 0$ to reduce the effect of Monte Carlo noise and aid in convergence.

## D.3 $\ell^1$-regularized coreset construction

Another option is to replace the cardinality constraint in Eq. (5) with the standard $\ell^1$-norm regularization popularized by the LASSO method [49] for sparse linear regression,

$$w^\star = \underset{w \in \mathbb{R}^N}{\arg\min} \quad \mathrm{D_{KL}}\left(\pi_w\|\pi\right) + \lambda \tilde{f}^T w \quad \text{s.t.} \quad w \geq 0, \qquad (74)$$

with regularization weight $\lambda > 0$ and potential scales $\tilde{f}_n = \mathrm{Var}_0\, f_n$. The potential scales $\tilde{f}$ account for the fact that the optimization is invariant to rescaling the potentials $f_n$ by positive constants; the optimization Eq. (74) is equivalent to optimizing $\mathrm{D_{KL}}\left(\pi_w\|\pi\right) + \lambda\|w\|_1$ with scale-invariant potentials $f_n/\sqrt{\mathrm{Var}_0\, f_n}$. We can solve this optimization for a particular value of $\lambda$ using proximal gradient descent,

$$w_{t+1} \leftarrow \mathrm{prox}_{\gamma_t \lambda}\left(w_t - \gamma_t \nabla \mathrm{D_{KL}}\left(\pi_{w_t}\|\pi\right)\right), \quad \mathrm{prox}_\lambda(x) := \mathrm{sgn}(x)\max\left(|x| - \lambda\tilde{f}, 0\right), \qquad (75)$$

where $\gamma_t = O(1/t)$ is the learning rate when optimizing based on Monte Carlo estimates of $\nabla \mathrm{D_{KL}}\left(\pi_w\|\pi\right)$. Although this approach generally provides less myopic solutions than greedy methods in the setting of sparse linear regression, there are two issues to address specific to sparse variational inference. First, since estimating the gradient of the objective in Eq. (74) involves sampling from $\pi_w$, the cost of iterations increases as $w$ becomes dense. To avoid incurring undue cost, a binary search procedure on the regularization $\lambda$ may be used. First, lower $\lambda_u$ and upper $\lambda_\ell$ bounds of $\lambda$ are initialized to 0 and $\max_n \left|\mathrm{Cov}_0\left[f_n, f^T 1\right]\right|$, respectively; these bounds ensure that $\|w\|_0 = 0$ when $\lambda = \lambda_u$ and $\|w\|_0 = N$ when $\lambda = \lambda_\ell$. Then in each binary search iteration optimization stage, keep track of $\|w\|_0$; if it ever becomes too large (e.g. $2M$), return early to prevent costly sampling steps.