[Reviews · NeurIPS 2019]

Reviewer 1



I thank the authors for the their clear response which solidified my view that this is a good contribution to NeurIPS In future work, I think it would be worth it to not only compare their method to normal coreset approaches but also to cheap-and-basic variational approximations like the Laplace approximation, if only to show me that I am wrong in suggesting that this could provide an alternative Best ----------------------------------------- The author(s) tackle the problem of Bayesian inference on large datasets. Coresets provide one approach to such solutions: simply sub-sample and reweight in a clever fashion the data so that inference on the sub-sampled data is almost the same as inference on the full dataset. They propose a variational inference approach which naturally builds a coreset-type approximation of the posterior. More precisely, they observe that the posterior is always in a (very high-dimensional) exponential family, with sufficient statistics the log-prior and the log-likelihoods of each datapoint. Thus, choosing an approximation which reweights each log-likelihood corresponds to seeking an approximation inside an exponential family, thus yielding computational tractability of the variational optimization of the KL divergence. In contrast, existing methods instead use Hilbert distances between the log-likelihoods. Their algorithm iteratively constructs a subset of the data by iterating the following two steps: 1. Select a new point to add to the subset. This is achieved by finding the minimizer among a random subset of the remaining datapoints of a stochastic approximation of the gradient of the KL divergence against the weights. 2. Recompute the weights of the log-likelihoods for all datapoints in the active subset. They propose a theoretical analysis of their algorithm and show that the exact version (no stochastic approximation of the gradient and no subsetting of the data in step 1 above) as a geometric approximation (Prop.1 and 2, line 200-208). They perform two tests of their method. The first is done on a Gaussian example, yielding closed-form updates. Their method is very good in this situation. The second is done on logistic and Poisson regression. This second comparison appears inconclusive to me and could be highly misleading due to the fact that it mixes together six different datasets. It is unclear to me whether the situation considered in this second experiment would be highly favorable to their method or whether it represents an honest test. I believe that this article is a strong accept but I am not completely certain in this opinion, due to my review time being limited. While the system flagged this submission as potentially similar to submission 1914, I am confident that there is zero overlap between the two submissions. Major remarks: - The algorithm has two stochastic steps: we’re sampling from the current approximation of the posterior, and we’re sub-sampling the potentials that we are willing to examine at each step. It would be useful to discuss the influence of the number of samples from the posterior-approximation and number of potentials on the precision of the algorithm. How should I choose these values in practice? What are their impact? Am I correct in stating that the guarantees of the algorithm only hold on the version of the algorithm which does not sub-sample the potentials and which examines all of them? - I have not understood where you discuss the impact of the speed parameter \gamma_t which appears in algorithm 1. Can you please discuss further how you choose its value and how its impact is felt? - Fig.2 does not feel very informative due to the fact that it mixes so many different sub-experiments. - This is more of a direction for future work, but one thing I’m curious about is how this method compares to Laplace approximations. In a large dataset, the Laplace approximation of the posterior distribution should be decent and should be easy to compute (or possibly at least easier to compute than the method you propose here). No minor remarks

Reviewer 2



# Strengths - Clear motivation and structure of the derivations. The paper and the introduced method are clearly motivated. The structure of the paper and the individual derivations is well structured and presented. - Some theoretic results from an information geometric viewpoint. The authors are able to show how their approach can be not only motivated by itself (Section 3), but also show how their approach can be nicels interpreted from an information gometry standpoint where they are able to show the nice result that the method optimizes a bound on the a symmetric version of the KL divergence. # Weaknesses - Seems more incremental than novel While the proposed model gives a new, clean approach on how to construct coresets, that is well motivated if we regard it in isolation, that benefit becomes a lot less clear if we take prior work on coresets into consideration. In that case it looks rather incremental and is lacking a strong motivation as to why it should be used compared to prior work in the area of coreset construction. - A weak experimental evaluation The method is only evaluated on a small set of datasets that are either synthetic or very low dimensional (D=10). Yet even there the authors can only demonstrate roughly similar performance to prior work, however at greatly increased cost. # Minor Typo In Equation (24), it should be \Sigma_w instead of \Sigma_p in the \mu_w computation unless I am mistaken. # Recommendation The work has potential, but there are still some weaknesses that should be addressed during the rebuttal, before I can recommend a clear accept POST-REBUTTAL: Having read the author response, I am convinced further about the significance of the contribution. Hence, I increase my score to 7.

Reviewer 3



Summary: -------- The paper proposes a new perspective on constructing Bayesian coresets. It is shown that the coresets are in fact in the exponential family. Based on that observation, the construction is formulated as an exponential family variational inference problem with a sparsity constraint. The optimization problem can be solved via greedy optimization and an interesting information-geometric view is provided. Review: ------- First of all, the paper is well written and the math seems to be sound. The derivations are easy to follow and authors do a good job communicating the main results. I find the exponential family view on the Bayesian coreset construction quite interesting. The greedy optimization method seems to be a "natural" way to solve the problem. As a second interesting contribution, I enjoyed the information geometric interpretation which gives a unifying view on coreset construction algorithms. I think that these theoretical insights are from interest to the community and could inspire new research in that field. However, the empirical evaluation seems to be quite limited. Based on the provided experiments it is hard to tell how significant this work is in practical terms. My concerns are: - The considered problems seem to be rather simple. They are more of a toy experiment nature (Gaussian toy data, logistic regression, Poisson regression). The significance of the experiments could be increased by considering more challenging (real-world) problems. - I have a couple of concerns about the plots in figure 3. - First of all, the plot (a) is hardly readable (too many lines in one plot). Also, there is no legend provided, which makes a direct comparison of the methods for one model impossible. - A more serious concern is that in plot (a), the GIGA optimization curves don't seem to be converged. Therefore, the claim that the proposed method achieves a better KL divergence using a smaller coreset size is not really justified. - In plot (c), why does the proposed method sometimes achieve very bad KL (points on the top)? - Additional to the KL plots, an experiment with a downstream task (i.e. MC sampling using the different coresets) would be interesting. Minor Comments: --------------- - Eq. 5: \pi_1 is used before it's actually defined - Is the fact that the KL divergence here is a Bregman div anywhere used? Conclusion: ----------- The proposed method is an interesting contribution and provides new insights on the construction of Bayesian coresets. However, in my opinion, the experiments don't show that the proposed method is superior in practice. First, the considered problems are quite simple. Second, the experiments seem to be immature and the significance is limited. I think the claims about the method are not justified by the experiments. By addressing the issues, the paper would have a much higher impact and would be suitable for acceptance at NeurIPS. I encourage the authors to improve the experiments and think the paper would then be a strong contribution. But in the current state, I tend to reject it.

[Author Response · NeurIPS 2019]

We thank the reviewers for their careful evaluation of our manuscript, and are glad to see that all reviewers appreciated the clear writing and value of our new unifying theoretical framework of coreset construction.

**R1 (# samples/potentials):** Two answers—one theoretical and one practical. On the theoretical side, we have new work that employs standard concentration inequalities to obtain the desired finite sample approximation error guarantees. This result will provide guidance on tuning $S$ in a follow-up paper, but is outside the scope of the present work. The practical answer is simpler: use as many features as is computationally feasible. We will add a note to the final draft regarding this point. Note that Props 1 & 2 are with respect to the true norms, and do not rely on any particular approximation/sampling scheme.

**R1 (step size):** $\gamma_t$ was not discussed adequately in the current draft; thanks for pointing this out. As is the case with many optimization algorithms, $\gamma_t$ is just a tuning parameter. Generally, it should decay such that the effect of tangent space approximation noise is eventually removed. We decided against developing Robbins-Monro-like theory for this due to both space constraints and the fact that in practice, decay rates other than $1/t$ worked best (see appendix D).

**R2 (novelty/significance):** We disagree that the work is incremental in view of past Bayesian coresets work, but we appreciate the point you raise, and will provide a more detailed discussion of the following points in the final draft. In particular, all prior Bayesian coreset constructions need a weighting distribution $\hat{\pi}$. Note that this is a very severe limitation; $\hat{\pi}$ isn't just a single tuning parameter, it's an entire distribution, and the fact that it is constant fundamentally limits the coreset construction (see results and discussion below right). Our first main contribution—which we believe is quite significant in the coresets literature—is to remove this bottleneck entirely. This demonstrates for the first time that fully automated, statistically rigorous coreset construction is possible. Further, past work provided no guidance on the meaning of $\hat{\pi}$; the second main contribution of the manuscript is a unifying info-geometric formulation that clarifies that $\hat{\pi}$ serves as the "anchor point" of a tangent space on the coreset manifold. We expect (and are already finding in our own ongoing work) that our new unifying info-geometric theory will open the door to many new Bayesian coreset-based inference methods. Naturally, given that our algorithm is the first instantiation of the new approach, we pay a computational price; we believe this price is well-worth it as a first foray into fully-automated coreset construction.

**R1&3 (unclear plots):** Thank you for your comments; in short, we agree completely. We tried to present results for numerous models / datasets / metrics despite limited space by combining results, but in hindsight "compressed" a bit too much. The two main metrics we want to use for comparison are computation time (i.e., construction cost) and coreset size (i.e., downstream inference cost). Thus we will (1) remove the iteration # plot (Fig. 3a), as it conflates these two metrics, and (2) split/format the remaining plots so that each dataset / model is clearly identifiable.

**R1,2,3 (experiments):** Although the underlying motivation for Bayesian coresets research is large-scale inference, the current work does not aim to explore the limits of data dimension and size in the new sparse VI formulation; this is a complex topic for which a statistically comprehensive treatment would merit a separate paper in itself (cf. Lucic et al 2018, "training Gaussian mixtures at scale via coresets" vs. many preceding papers on mixture coresets). Note that this is not unusual for a subfield still in its early exploratory phase (cf. the history of stochastic methods for regression or VI), where contributions tend to be foundational in nature, rather than computational. We focus here on the foundational problem of removing the weighting distribution $\hat{\pi}$ of past coreset methods; we verify that this is sound by demonstrating a reasonable level of performance on simple illustrative problems to avoid the confounding difficulties of more complex models. We will make these points clear in the final draft. However, to demonstrate feasibility in higher dimensions, we have increased the dimension of the synthetic Gaussian example to 200, and also applied our method to a new 301-dimensional regression problem with 300 Gaussian basis functions plus 1 constant function on 10,000 house sale records from the 2018 UK land registry dataset (both results shown right). This illustrates a key strength of our new methodology: previous Hilbert coresets eventually reach a performance limit due to using a single tangent space approximation, with quality depending on the choice of $\hat{\pi}$ (green,orange), while our method (blue) is "manifold aware" and continues to improve. To capture this in the final draft we will include the new regression experiment and increased-dimension Gaussian experiment, and increase the iteration count for all tests to clearly show the performance limit of Hilbert coresets in each. We will also highlight this limitation of Hilbert coresets in the text. Note: after submission we found that the Poisson regression log likelihood was numerically unstable in a rare circumstance that was triggered in one of the datasets (biketrips), leading to the anomalous result. We have fixed this instability.

**R2,3 (minor comments):** $\pi_1$ is defined in footnote 1, but we will emphasize next to Eq. (2). The Bregman comment will be removed as it is not used directly. We will address all typos. Thank you both very kindly for the careful edits!

[Meta-Review · NeurIPS 2019]

The reviewers agree that the paper offers an interesting technical development. I recommend that the authors take the reviewers' comments into consideration in revising this manuscript for the camera-ready.